



# Current structure, circulation and transport in the Central Baltic Sea observed by array of moorings and gliders

Taavi Liblik[1], Daniel Rak[2], Enriko Siht[1], Germo Väli[1], Johannes Karstensen[3], Laura Tuomi[4], Louise C. Biddle[5], Madis-Jaak Lilover[1], Māris Skudra[6], Michael Naumann[7], Urmas Lips[1], Volker Mohrholz[7]

[1] Tallinn University of Technology, Tallinn. 19086, Estonia

[2] Institute of Oceanology, Polish Academy of Sciences, Sopot, 81-712, Poland

[3] GEOMAR Helmholtz Centre for Ocean Research Kiel, Kiel, 24148, Germany

[4] Finnish Meteorological Institute, Helsinki, 00560, Finland

[5] Voice of the Ocean Foundation, Gothenburg, 42671, Sweden

[6] Latvian Institute of Aquatic Ecology, Riga, LV-1007, Latvia

[7] Leibniz Institute for Baltic Sea Research Warnemünde – IOW, Warnemünde, 18119, Germany

*Correspondence to*: Taavi Liblik (taavi.liblik@taltech.ee)

**Abstract.** An array of moored current meters and gliders were deployed across the Central Baltic Sea. Weak and low persistent currents driven by the wind-driven Ekman transport were observed in the upper layer in the offshore area. Stronger currents with higher persistency were observed near the coast, presumably driven by the local sea level gradient. The kinetic energy was largely observed to be in the low-frequency band (timescales >36 h), but higher kinetic energy on shorter timescales was observed in the areas, where the halocline was close to the seafloor. A strong and highly persistent gravity current was observed in the near bottom layer of the Fårö sill. Circulation pattern, current structure, and meridional transport varied on synoptic and seasonal timescale due to changes in local forcing and caused changes in the water column habitats. A low persistent, cyclonic gyre in the upper layer and meridional transport to the north in the deep layer prevailed in the seasonal (3-4 months) timescale. Circulation patterns with higher persistency formed under stable forcing. A cyclonic circulation in the upper layer was supported by southerly winds and reversed by northerly winds. Northward transport in the deep layer was intensified by northerly winds and reversed by southerlies. Due to the seasonality in the meridional wind component, the pattern associated with northerly (southerly) wind is more common in spring and summer (autumn and winter).

The deep-water transport is approximately 400 km$^3$ y$^{-1}$ towards the Northern Baltic Proper, which is almost 1/3 of the total deep-water volume in the Northern Baltic Proper, Gulf of Finland, and Western Gotland Basin. This transport mostly occurs during spring and summer and brings denser, oxygen-depleted water to the deep layer of these areas.



## 1 Introduction

The oceanic current field and circulation plays a crucial role in the redistribution of heat, salt, buoyancy and substances in the Baltic Sea and therefore has a large impact on ecological conditions in the sea. This study addresses the knowledge gap regarding the current structure and circulation in the Central Baltic Sea.

The semi-enclosed, shallow, brackish Baltic Sea has a strong, but variable vertical stratification that includes two pycnoclines: the seasonal thermocline and the permanent halocline. Stratification through the two pycnoclines impedes
vertical mixing, and the transport of substances between the layers is limited. Current profiling measurements in the Baltic Sea showed a strong link between the stratification and the vertical structure of currents (Bulczak et al., 2016; Suhhova et al., 2018). On the one hand, the pycnoclines determine the current shear maxima, but on the other hand, the current structure shapes the pycnoclines.

Thermohaline circulation, superimposed by the current field due to prevailing atmospheric forcing, determines the so-called
Baltic Sea Haline Conveyor (Döös et al., 2004), where saline water is transported from the North Sea towards the northeastern end of the Baltic (Liblik et al., 2018; Väli et al., 2013), salt is transported upwards by vertical mixing and other processes, such as upwelling and submesoscale flows (Lehmann et al., 2012; Salm et al., 2023; Stigebrandt, 2017), and outflow of the mix of fresh and salty water in the upper layer is transported towards the North Sea (She et al., 2007).

The Baltic Proper (Fig. 1, referred to as the Central Baltic hereafter) is the largest basin of the Baltic Sea. Several numerical
simulations have shown the cyclonic long-term mean circulation (Jędrasik and Kowalewski, 2019; Meier, 2007; Placke et al., 2018), which covers the whole water column (Placke et al., 2018) in the Central Baltic. The magnitude and persistency of the long-term flow are rather low (Jędrasik and Kowalewski, 2019; Liblik et al., 2022; Placke et al., 2018), likely due to high-temporal variability of currents under variable atmospheric forcing (Golenko and Golenko, 2012; Krayushkin et al., 2019). However, current fields with relatively high persistency and remarkably higher magnitude compared to long-term
mean occur during steady forcing conditions (Liblik et al., 2022). Simulations (Dargahi, 2019; Liblik et al., 2022) indicate these high-persistent features consist of several complex circulation cells, whose existence however has been verified with only limited observations at the eastern coast of the Central Baltic (Liblik et al., 2022).

Two quasi-permanent, sub-halocline circulations have been noted in the Central Baltic. A cyclonic gyre below the halocline in the Eastern Gotland Basin has been documented at the eastern flank of the basin based on moored observations (Hagen
and Feistel, 2004, 2007; Holtermann et al., 2014) and also from Argo float trajectories (Liblik et al., 2022). A quasi-steady northward-flowing gravity current is documented at the Fårö sill between the Fårö and Northern Deep (Liblik et al., 2022). According to model simulations, the gravity current has mean northward component of 10 cm s$^{-1}$ and it is suggested to play an important role in shaping the deep layer water properties in the Northern Baltic Proper and in the Gulf of Finland (Liblik et al., 2022). It also passes the Gdansk-Gotland sill (Krek et al., 2021). It is important to note that there is a lack of ocean
currents observations in the Central Baltic. Despite a few episodic and local time-series recording (Hagen and Feistel, 2004,



2007; Holtermann et al., 2014; Liblik et al., 2022) no wide spread and concurrent observations across all of the Central Baltic have been reported so far.

For estuarine systems, it has been shown that wind remarkably alters near-bottom currents and water properties for example for the Gulf of Finland (Liblik et al., 2013; Suhhova et al., 2018) and other such systems (Giddings and MacCready, 2017;
Pfeiffer-Herbert et al., 2015). Long-term ocean circulation model simulation indicated that halocline ventilation (meridional transport from south to north in the sub-halocline layer) correlate with low-pass filtered zonal wind stress on longer time scales (cutoff period 4 years, Väli et al. 2013). However, for the sub-halocline flow in the Central Baltic it is not clear how density-driven estuarine circulation and atmospheric forcing partition on shorter (annual and episodic) time scales.

Variability on shorter time scales (hours to days) is prominent in ocean current records as well as in sea level spectra and
various processes, such as internal waves, inertial oscillations, seiches, tides and (sub)mesoscale processes have been identified as drivers (Jönsson et al., 2008; Medvedev et al., 2013; Salm et al., 2023; Suhhova et al., 2018). Suhhova et al. (2018) showed that the spectral energy range mostly can be attributed to wind forcing and changes in ocean stratification. For instance, seiches prevailed in the current spectra in the period of strong winds and without seasonal stratification. Current spectra analysis based on the data of multiple simultaneous current-meters measurements is missing in the Baltic Sea
research.

Here we report on data from an international array of multiple acoustic current meters that was performed as Central Baltic Circulation Experiment (CABLE). Such an observing approach has been successfully applied to various open ocean and other estuarine systems (e.g. Fang et al., 2015; Hallock and Marmorino, 2002; de Jong et al., 2020; Lien et al., 2015; Shin et al., 2022), but to our knowledge not in the Baltic Sea so far. The moored observations of ocean currents are augmented by
underway glider measurements in the Fårö sill area and continuous measurements of water properties in the near bottom layer at three locations. The present study is the first general investigation of circulation and current structure based on the CABLE data but further studies, focused on various topics will follow.

This paper has four primary aims: The first aim is a description of the circulation features across the Central Baltic. The second aim is to verify and characterize the variability of deep layer circulation, particularly the current at the Fårö sill, by
in-situ measurements. The third aim is to describe the role of wind forcing on the sub-halocline currents and transport in the time scale from days to seasons. And finally, the fourth aim is to investigate the spatial distributions of current spectra during the periods with and without seasonal stratification.



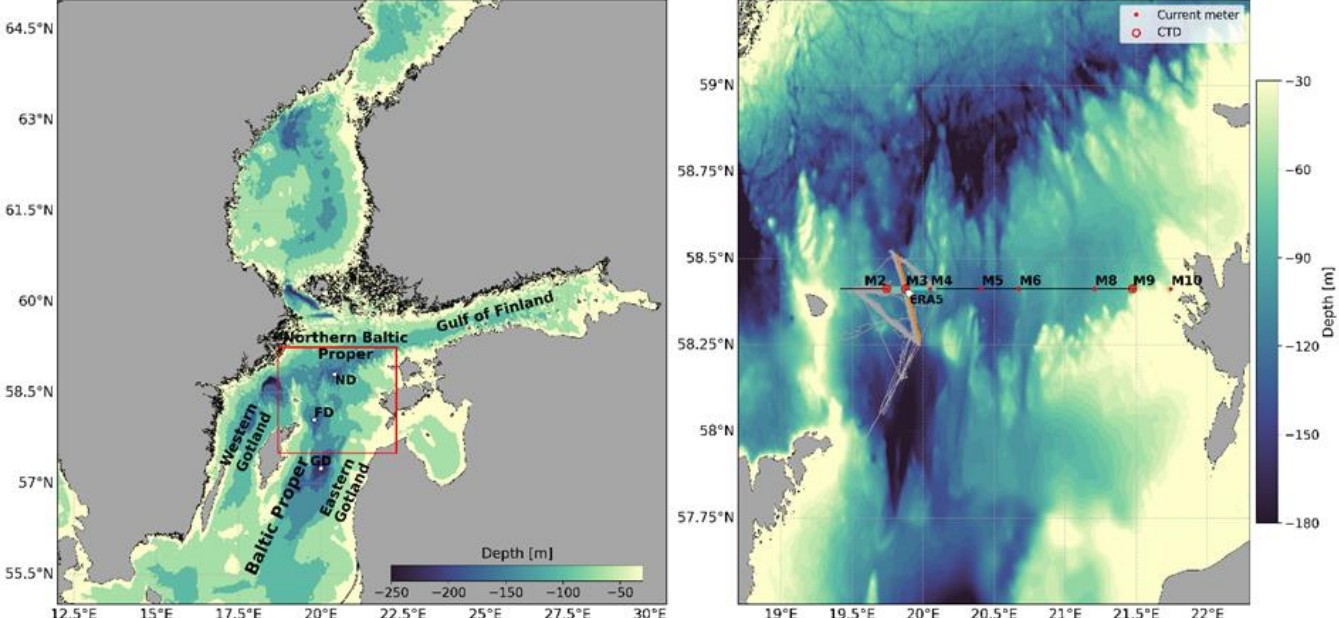

**Figure 1: Array of moorings carrying current meters (M2-M10) and path of glider surveys (grey) with bathymetry in the color scale (m). The black line represents the section, where transport calculations were done. The light blue dashed line between M2 and M4 is the zonal (across thalweg) glider section and the orange line is the meridional (along thalweg) glider section. Larger circles in M2, M3 and M9 represent near bottom CTD-recorded. The location of the center of ERA5 grid point is shown as white circle.**

## 2 Data and methods

The CABLE was carried out in the Central Baltic Sea using measurements by autonomous observational platforms.

### 2.1. Observations and data

An array of 9 moorings, 8 equipped with current profilers and one with point current meter, was deployed at the zonal section in 2022-2023. Eight out of nine moorings were recovered and recorded data successfully (Table 1). A key location in the section is the Fårö sill, where the mooring M3 was deployed. All other moorings were deployed to the same latitude as M3. At all locations, bottom-mounted current profilers (ADCP – acoustic Doppler current profiler, 300 kHz; Teledyne RDI) were used, except at station M10, where single-point model 106 current meter (Valeport Ltd) was deployed at 10 m depth. The sampling interval of ADCPs was 1 hour (average of 80 pings distributed over 1 h).





Velocities were measured with a vertical depth interval of 2 m starting from 4 m above the instruments. Due to different constructions of the moorings, the deepest available bin with data was 4 to 6 m above the sea floor. The vertical extent of available measurement bins varied from station to station (table 1).

Temperature and salinity time-series in the near bottom layer was recorded in the stations M2, M3 and M9 using Sea-Bird MicroCat recorders.

Glider surveys were conducted in the Fårö sill area to observe the water exchange and characteristics over the sill (Fig. 1). SeaExplorer gliders were deployed along the butterfly track between the stations M2, M3 and M4 (grey line, Fig. 1) from 11

April 2022 to 11 April 2023. This way data was recorded along as well as across the sill. The gliders performed a total of 14 missions collecting 23 026 vertical profiles in the region. Gliders profiled through the full water column, surfacing every 5 or 8 dives to receive a location fix and performing their inflections 2 m from the surface and 2 m from the seabed.

All gliders were equipped with an RBR Legato CTD sensor and a Nortek 1MHz AD2CP current profiler; as well as a suite

of basic biogeochemical sensors. The CTD recorded at 1 Hz, equivalent to approximately 10 cm vertical resolution while the ADCP collected a short vertical velocity profile (approx. 13 m effective range) every 5 seconds.

Hourly 10 m level wind velocities from ERA5 reanalysis data (Hersbach et al., 2020) for the period 1983 to 2022 (see Fig. 1 for location) were used in the analyses.


**Table 1. Metadata of stations. The M2 was deployed by RV E. M. Borgese, but recovered by RV Oceania. The M3 data was available up to 49 m depth until 14 January and up to 97 m depth afterwards.**

| Station | Deployment Period | Water depth (m) | Depth range of ADCP measurements (m) | Research vessel deployment & recovery cruises | Subhalocline thalweg orientation (respective from the north) |
|---|---|---|---|---|---|
| M2 | 09 May -06 Dec 2022 | 92 | 86 – 8 | RV Elisabeth Mann Borgese EMB293/ RV Oceania* | 300° |
| M3 | 09 May 2022 -14 Jan 2023 | 121 | 117 – 49 | RV Elisabeth Mann Borgese EMB293/EMB314 | 290° |
| | 14 Jan 2023 -21 Mar 2023 | 121 | 117 – 97 | | |





| M4 | 17 May 2022 - 22 Mar 2023 | 115 | 110 – 62 | RV Elisabeth Mann Borgese EMB293/EMB314 | 10° |
|---|---|---|---|---|---|
| M5 | 06 Apr -12 Oct 2022 | 131 | 127 - 67 | Aranda | 350° |
| M6 | 06 Apr -12 Oct 2022 | 101 | 97 – 9 | Aranda | 310° |
| M8 | 06 Apr -12 Oct 2022 | 92 | 89 - 9 | Aranda | 40° |
| M9 | 06 Apr -12 Oct 2022 | 69 | 63 – 7 | Aranda | - |
| M10 | 06 Apr -12 Oct 2022 | 35 | 10 | Aranda | - |

## 2.2. Data processing and calculations


The RBR Legato CTD was lag corrected following the Lueck and Picklo (1990) algorithms using coefficients provided by the manufacturers to minimise both thermal lag and thermal hysteresis.

Nortek 1MHz AD2CP data was processed using the Python gliderad2cp toolbox. Beam coordinate velocity profiles were

collected in 1 m bins with a 20 cm blanking distance every 5 seconds. Individual velocity measurements were regridded onto isobars to account for glider pitch variations, then coordinate transformed into east-north-up velocities. Full water column velocity profiles were calculated by referencing reconstituted shear profiles (using 2m vertical mean bins) to dive-averaged currents. Dive-averaged currents were determined using the Merckelbach et al. (2010) flight model, regressed with equal weighting against a vertical velocity constraint (determined from the pressure sensor) and a speed through water constraint

(measured as the mean velocity in the first 3 bins measured by the ADCP). Finally, shear-bias was corrected as per Todd et al. (2017). Final velocity profiles were validated by comparison to glider surface drift and ADCP bottom tracking.

All velocity data from moorings was recorded with 1h sampling interval. ADCPs used were programmed for a bin length of 2 m. The quality of the current velocity data was checked following the procedure developed by Book et al. (2007).


Mooring ADCP data was also transformed into parallel and perpendicular to subhalocline thalweg orientation, meaning following the topography of the deepest ridge (see table 1). The one-hour averaged time series was further low-pass filtered



36-h cutoff low-pass Butterworth filter. This way higher frequency oscillations, e.g. inertial oscillations, tidal and seiche-related currents, were suppressed. Similarly, band-pass filtering with relevant cut-off parameters was carried out to get signals of different frequency bands.

All velocity data was corrected for magnetic misalignment according to the IGRF-model (Alken et al., 2021).

We calculated the square of current velocity shear ($|\mathbf{S}|^2=(du/dz)^2+(dv/dz)^2$) using a two-sided derivative algorithm giving a value at a center point representing a six-meter layer.

The directional persistency parameter

$$P = \frac{\sqrt{u_m^2+v_m^2}}{\Sigma_i^n \sqrt{u_i^2+v_i^2}/n}, \tag{1}$$

the ratio of the vector mean speed and average flow speed (percentage) was used to characterize the flow tendency to be a vortex, back and forward, or unidirectional flow (Palmen, 1930). If the flow is a vortex or back-and-forth flow, the persistency parameter equals 0. In the case of unidirectional flow, the parameter is equal to 1. To estimate the correlation between the current and wind velocity vectors a complex correlation coefficient $\rho = R\ exp(i\alpha)$, introduced into physical oceanography by Kundu (1976), was used. Its magnitude $0 < \rho < 1$ measures the overall correlation of two series and the phase angle $\alpha$ displays the average counterclockwise angle of the current vector series with respect to the wind vector series.

For horizontal kinetic energy (HKE) spectral density investigations, the rotary spectral analysis technique (e.g., Emery and Thomson 2004) was applied to hourly average current data (HKE spectrum=Sp+Sn, where Sp and Sn are positive and negative rotary spectra).

To estimate the HKE of a given frequency band, we integrated the HKE spectral density in the corresponding frequency band.

We investigate the kinetic energy in three different frequency bands: broad semi-diurnal frequency band (BSD, ≤17 h), broad diurnal frequency band (BD, >17 h and ≤36 h) and low-frequency band (LF, > 36 h). The main contributors to the BSD are high-frequency currents, seiches, tides $S_2$ and $M_2$, inertial oscillations and to the BD seiches, and tides $K_1$ and $O_1$ (Jönsson et al., 2008; Lilover et al., 2011; Medvedev et al., 2013; Suhhova et al., 2018).





Salinity values are given as Absolute Salinity (g kg$^{-1}$) and calculated by TEOS toolbox (McDougall and Barker, 2011). Daily mean temperature and salinity profiles from glider data were calculated. Brunt-Väisälä frequency squared ($N^2$) was

calculated by TEOS toolbox (McDougall and Barker, 2011) and used to describe the strength of the stratification.

For transport and salt flux calculation, firstly the current velocities were interpolated to the same time grid with temporal spacing of 1 hour. Secondly, the spatial target grid with horizontal spacing of 60 m and vertical spacing of 1m from 70 m depth to the sea bottom was compiled based on EMODnet Digital Bathymetry (https://emodnet.ec.europa.eu/en/bathymetry,

downloaded on 22 April 2024) along the meridional transect of current observations (see dashed line in Fig. 1 for location)..
Thirdly, the current v-component values were linearly interpolated to the target spatial grid at every time-step. Fourthly, deepest values were extrapolated downwards, where necessary. Fifthly, values were extrapolated westward until boundary from M2 station. Sixthly, hourly and daily volume fluxes were calculated. Seventhly, salt flux was calculated on the basis of glider-time-series and volume flux. Linear regression between the near bottom meridional velocity at M3, and volume flux

($R^2 = 0.45$, p < 10$^{-10}$, n = 7345) and salt flux ($R^2 = 0.46$, p < 10$^{-10}$, n = 7345) in the section was calculated. The acquired relationships were applied to estimate the volume and salt flux in the section solely from the near bottom meridional velocity at M3. The effect of bottom friction through a logarithmic law of the wall with roughness length of 0.01 m was considered in the transport calculation.

Basin volumes were calculated according to the HELCOM subbasins borders (https://metadata.helcom.fi/geonetwork/srv/eng/catalog.search#/metadata/e5a59af9-c244-4069-9752-be3acc5dabed, downloaded on 22 April 2024) and EMODnet Digital bathymetry (EMODnet Bathymetry Consortium, n.d.).

The four periods with duration of 5-7 days were chosen to analyze the current structure during different atmospheric forcing and hydrographic situations. The periods were carefully selected to have periods with stable forcing. Periods were characterized by high P (79-97%) of wind.

## 3 Results

### 3.1. Time-series of forcing, hydrography and currents

The time series of wind (Fig. 2a) and water column structure (Fig. 2b-d) from April 2022 to March 2023 revealed a typical seasonal cycle for the area under discussion. Strong winds were more frequent, and winds from the southwest prevailed more in autumn and winter. The wind was generally weaker, strong wind events were rarer, and southwesterly winds did not



prevail as much in spring and summer (Fig. 2a). Daily cumulative wind stress in August-September shifted to the southwest,
being opposite to the prevailing northeast direction (in the case of southwesterly winds).

Three water masses separated by the two pycnoclines (seasonal thermocline and halocline) were observed during the seasonally stratified period in the area: warm upper mixed layer from the sea surface to 10-20 m depth, cold intermediate layer at 40-60 m depth, the deep layer below 80 m depth.

Seasonal stratification formed in spring and was at its peak in August when the sea surface temperature exceeded 20 °C (Fig.
2b). Seasonal stratification was maintained not only by the temperature distribution but also by salinity – the water was fresher in the upper mixed layer than below the thermocline (Fig. 2c). Convection and wind mixing started to erode the seasonal pycnocline in the second half of August and, as a result, the upper mixed layer deepened from 10-20 m in summer to 50-60 m in January, which is well visible in the $N^2$ time-series (Fig. 2d). The halocline was located mostly between 60-80 depth, and it had slightly shallower position in July-August and in January (Fig. 2d).

The developments in wind forcing and stratification are well reflected in the time series of currents (Fig. 3) and its shear (Fig. 4). Mostly, a three-layer current structure was present during the period with seasonal stratification. The vertical location of the current shear maxima was linked to the pycnoclines. The current was stronger near the coastal boundaries and weaker offshore. The latter was also reflected in the time series of shear i.e., the shear was weaker at offshore stations. The meridional current component above the halocline was predominantly negative (flow was to the south) at the western side
(station M2) of the section while northward flow was more frequent at the eastern side (M9, M10).







**Figure 2: Time series of wind stress ($\tau_x$, $\tau_y$) and cumulative wind stress ($\tau_{xc}$, $\tau_{yc}$, solid lines) from ERA; temperature, salinity, and Brunt-Väisälä frequency squared ($N^2$) from gliders from 6 April 2022 to 21 March 2023. Dashed lines in panel a represent the climatological mean cumulative wind stress from 6 April to 21 March in the period from 1983 to 2022. Temperature and salinity measurements are conducted in the Fårö sill area (see glider path in Fig. 1).**






**Figure 3: Time series of eastward (u) and northward (v) velocity components from 6 April 2022 to 21 March 2023. All ADCPs were on bottom frames and the first measurement level was centered from 4 to 6 m from the sea floor.**





**Figure 4: Current shear squared ($|\mathbb{S}|^2$) at seven stations from 6 April 2022 to 21 March 2023.**



**3.2. Mean patterns and characteristics**

Mean profiles of current speed, persistency P, and velocity during the seasonally stratified period (from 18 May 2022 to 11 October 2022) and in the period without seasonal stratification (from 01 January 2023 to 20 March 2023) are shown in Figs. 5-6. All parameters reveal a pronounced vertical structure.

We first investigate mean peed and persistency P during the seasonal stratification. The stations (M2, M6, M8, M9, M10), where acoustic measurement cells reached the surface layer, had the highest mean speed in the topmost layer, 9.5-12.1 cm s$^{-1}$ (Fig. 5a). The mean speed in the upper layer was strongest at M9, indicating that the strongest currents occurred near the basin boundary (coast). It has to be noted that M10, which was even closer to the boundary, had a gap in data and does not fully cover the period from 18 May to 11 October 2022. Observations revealed very low persistency in the upper layer, with values below 30% at all 5 stations and only 10-13% at M6 and M8 (Fig. 5b).

The mean speed structure at M2, M6, and M8 had a minimum at 20-30 m depth (i.e. around seasonal thermocline) and a maximum at 50-60 m depth, which is just above the halocline. Such a structure was not revealed at M9. The M9 velocity field had only a slight minima-maximum structure at 15-20 m depth and a steep drop from 12 cm s$^{-1}$ at 20 m to 4 cm s$^{-1}$ at 62 m.

The mean speed decreased through the halocline while P rather increased beneath the halocline. The vertical maximum of P was just below the halocline at all deep enough stations except M3. The near-bottom layer at the Fårö sill station M3 had the highest mean speed (14-15 cm s$^{-1}$) and persistency (P = 76%) of the whole measurement campaign. High persistency was also found at 100-105 m depth in M4 (P = 61%) and 58 m in M9 (P = 61%). The prominent persistency maximum at the thermocline was also at M9 (P = 51%).

Observations in the halocline and deep layer of the two stations (M3, M4) during the period without stratification reveal higher mean speeds and lower persistency compared to the summer. This probably indicates the impact of stronger winds, mostly from the southwesterly direction in winter (Fig. 2), which on one hand induce stronger currents, but on the other hand, reversal events of the persistent gravity current towards the north were more frequent, and therefore P decreased.

The mean velocity profiles during the seasonally stratified period reflect the cyclonic circulation in the upper layer and in the cold intermediate layer (Fig. 6). The mean flow was stronger at the interior basin and weaker in the central part. Mean flow in the deep layer differed from the layers above in most of the stations. Despite steering due to the local bathymetry, the meridional component of the mean flow was to the north at almost all stations. The strongest mean flow occurred in the Fårö sill (M3). An opposite flow towards the south occurred at M2.

The mean flow structure below the halocline in the seasonally non-stratified period at M3 and M4 was very similar to the ones during seasonal stratification. The mean velocity was a bit weaker, although the mean speed was higher in the winter period. This is another sign of the impeding impact of the southwesterlies on the northward gravity current.







**Figure 5: Profiles of mean current speed and persistency during the period of seasonal stratification, from 18 May to 11 October 2022, and during the period without seasonal stratification, from 1 January to 21 March 2023.**




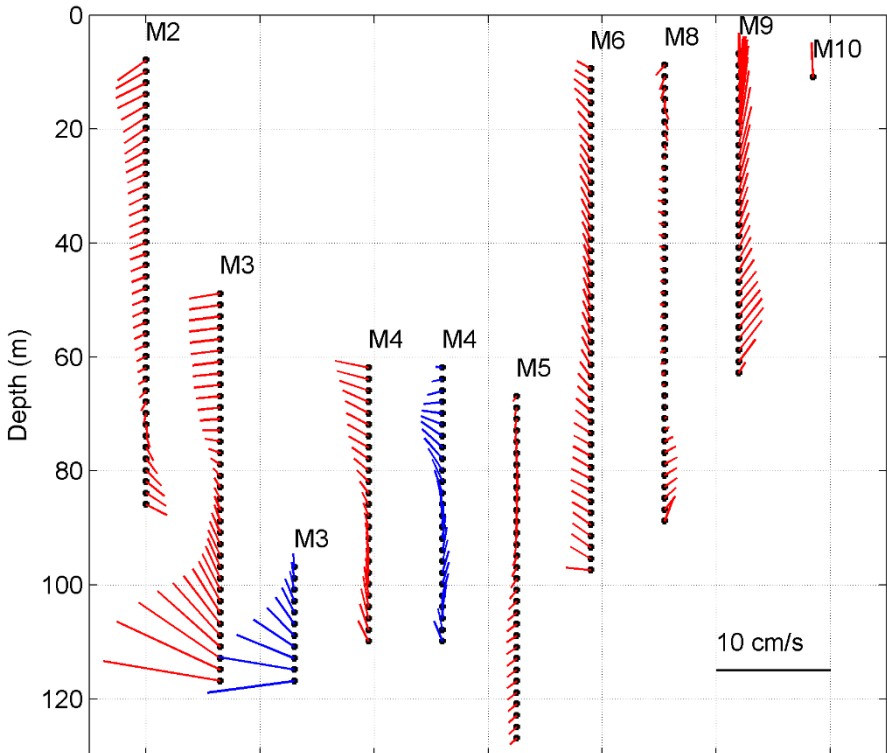

**Figure 6: Mean profiles of current velocity during the period of seasonal stratification, from 18 May to 11 October 2022 (red), and during the period without seasonal stratification, from 1 January to 21 March 2023 (blue). The seafloor was 4-6 m below the deepest bins.**


The vertical distribution of kinetic energy spectra show a clear maximum at all stations at the energy band from 12 to 17 h, with the most prominent peak at 14 h, corresponding to the inertial wave period (Fig. 7). The energy was highest in the upper layer (>100 (cm s$^{-1}$)$^2$) and decreasing downwards. Both the thermocline and the halocline likely dissipated and reflected the energy and therefore the energy levels in the cold intermediate layer and deep layer were lower. Smaller elevated energy

peaks could be noted from the 17 to 36 h range, e.g., the peak around 24 h at M2 during the stratified period corresponds to the diurnal tidal constituents. Higher energy at low frequencies (> 36 h) is evident, particularly at the stations near the coastal slope (M8, M9) and in the deep layer at the Fårö sill (M3), especially during the period without seasonal stratification. Low frequency currents were weakest in the middle of the basin.

Next, we describe the vertical distribution of integrated kinetic energy among three bands (Fig. 8): BSD (≤ 17 h); BD (> 17 h

and ≤ 36 h); LF (> 36 h). The highest amount of energy was in the LF band. Most of the stations and depth ranges had LF



contribution higher than 70% of the total kinetic energy budget. Particularly high, mostly > 80%, was the LF component in the cold intermediate layer and in the deep layer. The LF was especially high in the deep layer at the Fårö sill (M3) in the period without seasonal stratification. The BD share in the Fårö sill was also elevated compared to other stations, but the LF strongly prevailed there, as > 90% of the energy was in the LF band both with and without seasonal stratification. The

energy at the LF band was also high in the upper layer at the coastal slope (M9).

There was elevated energy at BSD and BD in the upper layer (10-20 (cm s$^{-1}$)$^2$), while it was about three-fold lower below the thermocline. The exception was the station M9 at the coastal slope, where such a decrease did not occur downwards. The higher energy at the BSD and BD was revealed at the stations where sea depth was in the range of the halocline depth (M2, M9). Therefore, LF contribution was relatively low in the energy budget near the bottom in these areas, namely, 40% at M9

and 53% at M2. The higher energy at the BSD was revealed also in the 60-70 m layer at M4 during the period without seasonal stratification. Likely, this is related to the absence of the thermocline, which allowed wind-generated motions such as inertial waves to penetrate deeper compared to the summer period.





**Figure 7: Vertical distributions of HKE spectra (HKE spectral density multiplied by the frequency) during the period of seasonal stratification, from 18 May to 11 October 2022 and during the period without seasonal stratification, from 1 January to 21 March 2023.**





**Figure 8: Mean HKE (left) and its share (right) within three frequency bands during the period of seasonal stratification, from 18 May to 11 October 2022 and during the period without seasonal stratification, from 1 January to 21 March 2023. The extent of the x-axis in the panel e is different from the panels a and c.**

## 3.3. The role of wind

The role of wind on current velocities was analyzed in terms of complex correlation (Kundu, 1976). The correlation strength $\rho$ and angle $\alpha$ between the low-passed wind and current velocities were calculated with different cut-off periods (36h, 5d, 10d; Fig. 9). The changes of $\rho$ in the case of different time-lags in the range of cut-off periods were calculated. Although slightly higher $\rho$ values occurred, when a time-lag applied compared to the zero time-lag, the changes were small. We present the $\rho$ and $\alpha$ for the time lags corresponding to half of the cut-off period i.e. time lags were 18 h, 2.5 d, 5 d (Fig. 9).



Rather low ρ at most of the stations and layers were found with 36 h cut-off period during a seasonally stratified period. Higher ρ was found in the upper layer of the three stations (M2, M6, M9; ρ = 0.35-0.44), and at M10 (ρ = 0.52, not shown in Fig 9). The lowest ρ in the upper layer was found at M8 (ρ = 0.25). The angle between wind and current in the upper layer (α) corresponded to the Ekman theory, i.e., the current vector was to the right from the wind vector. Veering was less pronounced near the eastern coast at stations M9 and M10, as the current was there likely steered by the coastal boundary.


The correlation ρ was very low in the cold intermediate layer, except at M8 and M9, where slightly higher values (ρ = 0.30-0.35) were observed. A slightly elevated correlation was between wind and upwind current in the near-bottom layer at M8 and M9. A remarkably higher correlation (ρ = 0.54-0.58) was found between wind and upwind current for the deep layer of M3 and M4 during the period without seasonal stratification.


A considerably higher correlation between wind and current could be found, particularly in the deep layer, when 5 d and 10 d filtering is used. Despite local modifications of deep layer currents due to morphology, the main pattern was similar among most of the stations. Positive meridional component of deep layer current correlated with winds from the northerly sector. The highest ρ (0.81) was found in the deep layer of the Fårö sill (M3) in winter.






**Figure 9: The mean angles between current direction and wind direction (grey vectors), and ρ (color scaled dots) between wind and current velocities during the period of seasonal stratification, from 18 May to 11 October 2022 and during the period without seasonal stratification, from 1 January to 20 March 2023. The angle between wind vectors and current vectors is the phase angle α according to Kundu (1976). The zero angle is oriented upwards as illustrates the red vector in panel a. The relationship between wind and current was calculated for low-passed (36 h, 5 d, 10 d) velocities. Time lags between wind and current 18 h, 2.5 d, and 5 d, respectively, were taken into account.**

## 3.4. Quasi-steady circulation patterns

Next, we analyze the current field during quasi-steady forcing (Fig. 10). The four periods were selected to display the reaction of the current structure during different atmospheric forcing and hydrographic situations: 1) 20 July – 24 July 2022; 2) 29 August – 4 September 2022; 3) 10-15 October 2022; 4) 10-16 January 2023. The first two were under strong seasonal stratification conditions, whereas southwesterly wind prevailed during the first period and northeasterly wind during the





second period. Southwesterlies also prevailed during the third and fourth periods, but seasonal stratification had started to erode and upper mixed layer deepened in the third period and the thermocline had vanished in the fourth period (Fig. 2). The

mean resultant wind vectors of each period (Fig. 10b) showed persistency of 79-97% during the selected periods.

Only data from M3 and M4 were available from the fourth period, and only data from the second and the third period from M10. Unlike the overall low persistency of currents during seasonally stratified and non-stratified periods (Fig. 5), the P was rather high and mostly over 50% during the periods. Moreover, P was over 70% in all periods/stations, at least in some parts of the water column.

The first period during the southwesterly wind impulse was characterized by basin-wide cyclonic circulation in the upper layer, which could be seen as a high persistent current towards the north at M8 (P = 85-86%) and M9 (P = 88-94%) on the eastern side of the section and towards the west at M2 (P = 70-79%). The flow was to the east in the upper layer at M6 (P = 50-65 %). Mean velocity in the cold intermediate layer was towards the east at M2 (P = 62-72 %) and M6 (P = 60-75 %) and towards the west in the near bottom layer near the eastern boundary at M8 (P =20-70%) and M9 (P = 20-80%). Latter

westward (offshore) flow was probably part of a downwelling cell. The P at M8 and M9 decreased downwards, and the low values were probably related to the downward movement of the thermocline during the downwelling process. The flow was highly persistent (P = 70-95%) and to southerly directions in the deep layer, except at M4 and M5. The flow was towards the northwest at M5 (P= 75-85%), while it was variable and low persistent at M4 (20-40%).

The third period had similar atmospheric forcing as the first period, i.e. southwesterly winds prevailed, but the seasonal

thermocline was weaker and deeper. A similar cyclonic circulation structure revealed as highly persistent current towards north and northeast in the upper layer and cold intermediate layer at M9 and M10 (P = 97-99 %) and at M8 (70-90%), and current towards southwest at M2 (72-90%). A relatively weak but persistent current towards the north in the middle of the section at M6 (P = 75-83%) indicates the possible existence of stable meanders within the basin-wide cyclonic circulation. The flow was less persistent compared to the first period, but mostly to southerly directions in the deep layer.

There was a tendency for weaker persistency in the deep layer and stronger in the cold intermediate layer during the third period, while it was the opposite during the first period. Despite the latter, persistency in the deep layer of the Fårö sill (M3) was still quite high (P = 70-87%).

The current structure at M3 and M4 during the fourth period without seasonal stratification was very similar to the third period, but deep-layer currents were stronger, and persistency was higher there.

Northerly-northeasterly winds prevailed during the second period and caused along-shore flow towards the south (M10, P= 65 %; M9, P = 80 %) and southwest or west (M8, P = 80 %; M6, P = 45%; M2 P= 47-56%) in the upper layer at the eastern coast. The westward advection in the upper layer was compensated by a strong and persistent current towards the east in the intermediate layer and deep layer at M8 and M9 (P = 70-96%). This flow pattern displays the upwelling circulation scheme along the eastern coast. One could note that velocity vectors and also persistency (P = 94-96%) were very similar below 60

m depth at M8 and in the depth range of 40-60 m at M9. The latter is likely a sign of halocline uplift and advection of the deep waters to shallower depths. Mostly, strong and highly persistent flow towards northerly directions was observed in the





deep layer (P = 90-99%). The exception was M2, where the opposite flow was observed in the near-bottom layer (P = 76-80%), and M5, where a moderately persistent (P = 48-50%) weak eastward flow was observed.

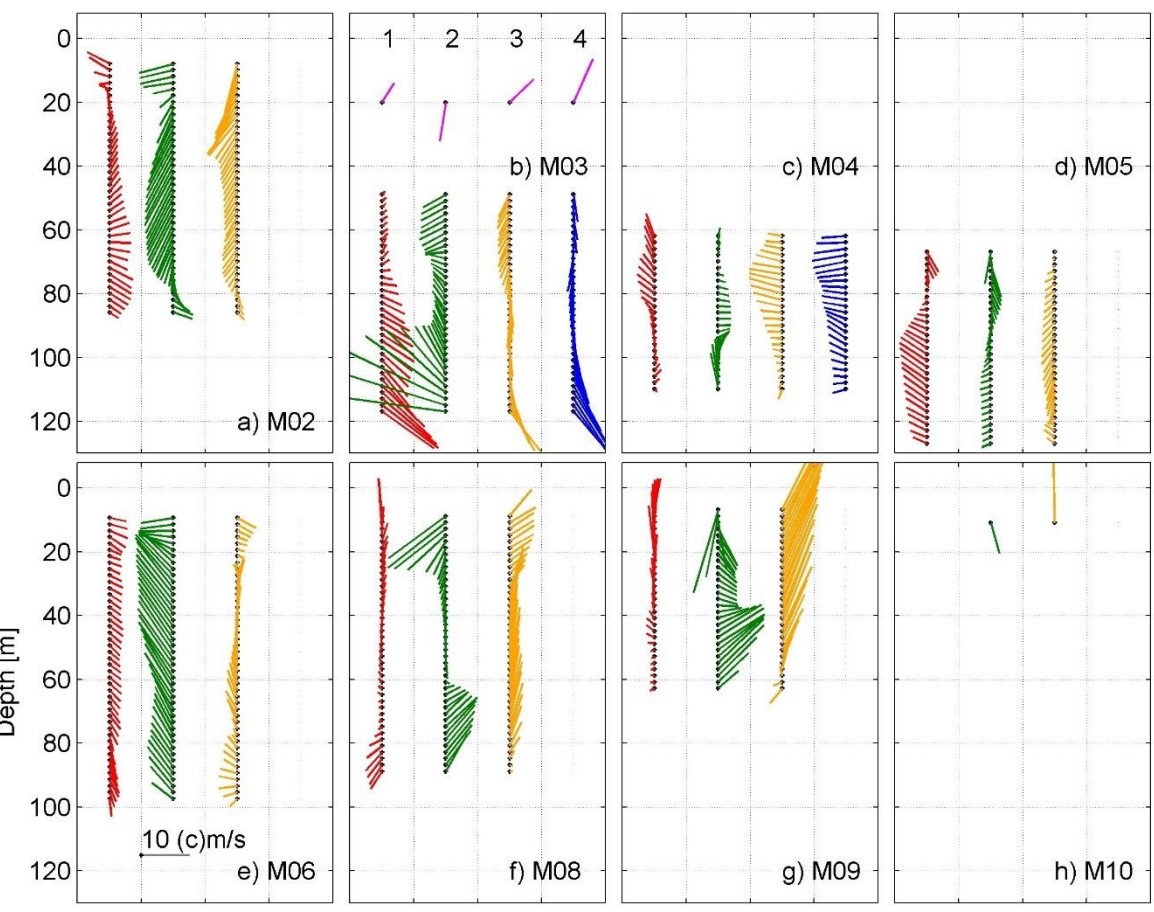

**Figure 10. The mean resultant wind vectors (panel b, pink lines) and mean profiles of current velocity vectors during four periods: 20 July – 24 July 2022 (red), 29 August – 4 September 2022 (green), 10-15 October 2022 (orange) and 10-16 January 2023 (blue). The scale is shown in panel e and corresponds to 10 cm s⁻¹ and 10 m s⁻¹ for the current and wind velocity, respectively.**

**3.5. Current and transport in the deep layer**

Near-bottom velocities below halocline reveal quite high variability, but the prevailing meridional current component was positive in most of the sections, except at stations M2 and M5 (Fig. 11a). By far, the strongest flow occurred at Fårö sill (M3). Fårö sill is a narrow and relatively shallow passage between wider deep areas in the south and north. When approaching from the south towards the sill, the cross-sectional area of the flow decreases rapidly. This narrowing is partly



compensated by the increase in the current velocity. However, the predominant southward flow at M2 at the western flank of
the sill suggests that not all the volume flux passes the sill, but a portion of it returns and moves back southward.

M5 was located in the deep valley (Fig. 1) and the near-bottom current was very weak there (Fig. 5).

On one hand, considerably high ρ was noted between current in the deep layer at stations M3, M4, M6, and M8. On the other hand, the deep layer flow towards the north at these stations was significantly correlated to the wind from northerly directions (Fig. 9). As a result, one could see two events created by southwesterly wind impulses (July and late September –
early October) when the meridional current was directed southward at all mentioned stations (Fig. 11a). Decreases in salinity and temperature at the sill were observed during these reversal events (M3, Fig. 11b). At the same time, lower salinity and higher temperature were observed in the near-bottom layer at M9, which is a sign of downwelling (Fig. 11b). Often, stronger currents also simultaneously occurred at these four stations, although M3 stood out as the one with remarkably higher magnitude compared to the other three. During these events, salinity and temperature were higher at the sill (M3, Fig. 11b)
and in the near-bottom layer at the eastern coast (M9, Fig. 11b).

The daily meridional transports below the halocline (70 m is considered as the upper boundary) in the section (Fig. 1) varied in the order of 1 to 10 km$^3$ d$^{-1}$ (Fig. 11c). Despite short-term reversals, the cumulative meridional transport towards north was approximately 270 km$^3$ from mid-May to mid-September 2022 while it was slightly negative from October 2022 to mid-January 2023, and 140 km$^3$ from mid-January to mid-March 2023. These volume fluxes were in accordance with cumulative
meridional wind stress, which was approximately 1 N m$^{-2}$ d$^{-1}$, 6 N m$^{-2}$ d$^{-1}$, and -1 m$^{-2}$ d$^{-1}$, respectively during the same periods (Fig. 2a). Thus, during the periods (seasons), when meridional wind stress was southward or close to neutral, the meridional transport towards the north was approximately 70 km$^3$ month$^{-1}$. The mean meridional cumulative wind stress in 1983-2022 has been in the same order as during the study period (Fig. 2a).

Our observations did not cover April and May, but from the wind regime, we could assume the total volume flux below 70 m
from mid-May 2022 to mid-May 2023 was in the order of 400 km$^3$ y$^{-1}$. The volume of the water below 70 m in the Northern Baltic Proper and the Gulf of Finland (according to the HELCOM borders between subbasins in the Baltic, https://helcom.fi/baltic-sea-trends/data-maps/, accessed 24 April 2024) is 740 km$^3$. However, not all the subhalocline water stays in the Northern Baltic Proper and/or arrives in the Gulf of Finland. Some of it flows around the Gotland Island to the Western Gotland basin, which has a volume of 705 km$^3$ below 70 m depth. Thus, the volume of the whole marine area,
which marine boundaries are Aland sill and Archipelago Sea in the north and Hoburg-Midsjö banks in the south of the Western Gotland basin (Leppäranta and Myrberg, 2009), is 1445 km$^3$. According to our estimates, nearly one-third of the deep layer (>70 m) water mass in this area was renewed annually.

Similar tendencies occurred in the salt flux, which varied mostly in the range of 100-1000 t s$^{-1}$ (Fig. 11d). Using the same approach as for transport, the annual salt transport would be approximately 4.5 • 10$^9$ t, which advected to the area with a rate
of 7 • 10$^8$ t month$^{-1}$ during the periods when meridional wind stress was southward or close to zero, i.e. from May to September 2022 and from mid-January to May 2023. Net meridional salt flux was negative below 70 m depth from October to mid-January.



The mean salinity and temperature distributions derived from glider measurements across and along the thalweg of the sill from April 2022 to March 2023 shown in Fig. 12 (a, d, g, j) illustrate the water exchange process between the Fårö Deep and

Northern Deep. The zonal transects reveal the structure of the gravity current. Isopycnals (shown only in panel d), which follow isohalines, were elevated on the eastern flank of the sill. The meridional transect illustrates the blocking effect of the sill. The water was clearly warmer and saltier in the south from the sill. Likewise, it displays the intrusion of more saline and denser water to the Northern Baltic Proper. Thus, this transport increases stratification in the downstream basins.

As revealed from near-bottom time-series (Fig. 11b), salinity and temperature are strongly linked to the flow direction at the

sill. Distributions during the active overflow towards north (Fig. 12b, e, h, k) and during the reversal event (Fig. 12c, f, i, l) display the spatial view of processes behind this temporal variability.

The across-thalweg salinity and temperature gradients were especially strong during the active overflow process (Fig. 12b, e). Warmer and saline water flowed towards north along the eastern flank of the sill. The meridional view (Fig. 12h, k) shows how the denser water moving towards the sill has penetrated about 20-30 m upwards from its vertical position in the

south. Denser water subsided along the trench deeper on the other side of the sill.

No clear gravity current structure of isolines across the thalweg could be seen during the reversal event (Fig. 12c, f). Isolines are squeezed together and pressed downwards in the south from the sill and overflow of the warmer and saltier water did not occur (Fig. 12i, l).

The glider and ADCP data at the sill area have several other details to be studied in further studies. In the present paper, we

do not go further with these.







**Fig 11. Current velocity along the local thalwegs (see Table 1) in the near-bottom layer (a). Temperature and salinity at 120 m depth at station M3 at Fårö sill, at 89 m depth at M2 and at 67 m depth at M9 (b). Meridional volume transport and cumulative transport (c), and meridional salt flux and cumulative salt flux (d) in the layer below 70 m through the zonal section (see dashed line in Fig. 1). Transport and flux marked by the dashed line in panels c and d show the time-series derived from the M3 velocity (see data and methods for details) while solid lines represent time-series where the whole dataset (glider data; ADCPs at M2, M3, M4, M5, M6, M8) is included. All time series are low-pass (36 h) filtered.**





**Fig. 12. Temperature (°C) and salinity (g kg⁻¹) distributions derived from glider measurements across and along the thalweg of the Fårö sill (Fig. 1). Mean distributions across (a, d) and along the thalweg (g, j) from April 2022 to March 2023. Distributions across (b, e) and along the thalweg (h, k) during the overflow event from 28 to 30 August 2022 and from 31 August to 1 September 2022, respectively; and across (c, f) and along the thalweg (i, l) during the reversal event from 6 to 8 January 2023 and from 15 to 18 January 2023, respectively. Black lines represent temperature and salinity isolines with step of 0.1 g kg⁻¹ and 0.1 °C, respectively. White dashed lines in panel d represent potential density isolines, which almost fit with isohalines.**




## 4. Discussion

Simultaneous measurements with the array of 8 current meters, continuous glider survey and near-bottom measurements of
water properties were used to analyze the current structure from hourly to seasonal time scales in the Central Baltic Sea.

The mean circulation in the upper layer in summer was cyclonic, as shown from numerous ocean model simulation data
analyses (e.g. Jedrasik et al., 2008; Meier, 2007). The persistency of this mean circulation was very low. The prevailing
current, volume transport and salt flux towards north below the halocline were detected. This is a lower limb of the estuarine
circulation (Elken et al., 2003; Giddings and MacCready, 2017; Liblik et al., 2013), which appears as a gravity current
(Liblik et al., 2022; Lilover et al., 1998; Zhurbas et al., 2012) in the overflow regions.

According to Meier (2007), the mean circulation below halocline appears as a basin-wide cyclonic gyre in the Northern
Baltic Proper, which is enhanced (stronger currents) in winter (Liblik et al., 2022). Our observations rather showed
prevailing flow towards the north in the deep layer in the whole section except the westernmost part (M2), where the
dominating flow direction was to the south. The spatial mean current velocity field in the layer below 70 m depth derived
from the glider mounted ADCP reveal the pattern of flow towards north along the deepest part of the sill and southward flow
in the eastern flank of the sill (Fig. 13). We cannot confirm whether the observation of this southward flow is a signature of
the southward limb of the basin-scale gyre or a local feature in the western flank of the Fårö sill. High-resolution simulation,
validated with measurements presented in this study, should describe the spatial extent and patterns of the feature.

The mean current structure at offshore stations revealed a minimum of around 20-30 m depth (i.e. around seasonal
thermocline). Such a structure was not observed near the coast. The vector correlation between wind and current velocity
also indicated a distinction between offshore and coastal areas. The angle between wind and current corresponded to the
Ekman current in the upper layer at the offshore stations (e.g. Lass et al., 2001; Lilover et al., 2011) while the sea level
gradient and boundary rather drove and steered the current near the coast. The two observations of distinction suggest that
the Ekman spiral was more pronounced in the offshore area while geostrophic current rather prevailed near the coast due to
the sea level gradient.
The strongest current with the highest persistency of the whole experiment was observed in the near bottom layer of the Fårö
sill. The observed behavior of this current largely agrees with the earlier suggestions given by simulation (Liblik et al.,
2022).



The wind stress along the axis of the Baltic Proper, i.e., from a southerly or northerly direction, and changes in stratification considerably altered the mean circulation. The flow patterns during quasi-steady forcing and stratification conditions had
higher persistency and differed considerably from the mean circulation scheme.

Cyclonic circulation was stimulated by southerly and westerly winds, which caused a strong northward current and downwelling along the eastern coast and a slowdown or even reversal of the gravity current in the deep layer. Such circulation pattern is more common in autumn and winter due to seasonal changes in the wind field (Soomere and Keevallik,
2001). The stronger current near the eastern coast was a result of Ekman transport-driven convergence, which in turn caused a sea level and pressure gradient and resulted in a geostrophic flow. The more stable flow, which was in geostrophic balance, might be the reason why higher energy at LF (> 36 h) was detected in the stations near the coast (M8, M9). Another reason might be the existence of coastal trapped waves that occur in the time scale of a few (2-3) days (Lass and Talpsepp, 1993; Pizarro and Shaffer, 1998; Talpsepp, 2006).

Strong southward flow along the eastern coast and westward flow at offshore stations were observed during northerly wind domination. Compensating onshore flow in the cold intermediate layer and deep layer, and uplift of the halocline along the eastern coast were observed while the southward volume flux in the upper layer was meridionally compensated by prevailing northward flow in the deep layer, i.e. the gravity current in the deep layer was encouraged. Such an acceleration of the deep
layer current in the Fårö sill was suggested by simulation (Liblik et al., 2022) and a similar structure has been observed in the Western Gotland Basin (Shaffer, 1979) and in the Gulf of Finland (Liblik et al., 2013; Lilover et al., 2017; Lips et al., 2017). Such wind-driven circulation pattern is more common in spring and summer.

These flow patterns could drastically alter water column habitat at the coastal slopes and fluxes between sub-basins. Besides
well-documented consequences of the upwelling process on the sea surface (e.g. Lass et al., 2010), the subsurface circulation could cause vertical movements of the halocline up to 30 m (Liblik et al., 2022). As a result, we observed the changes in the near-bottom salinity with the amplitude of 2.5 g kg$^{-1}$ at M9 (Fig. 11b). Since the halocline is well correlated to the oxycline (Liblik et al., 2018; Meyer et al., 2018; Rolff et al., 2022), one could expect that during northerly winds, upward intrusion of oxygen-depleted water along the eastern coast occurred. Our shipborne CTD profiling (not shown) during the cruises in
April and October 2022 revealed that hypoxia (2.9 mg l$^{-1}$) started from the salinity of 8.5-9.0 g kg$^{-1}$, i.e. the near-bottom layer of M9 certainly experienced periods of high oxygen conditions, but also synoptic events of hypoxia occurred due to upward excursions of oxygen-depleted deep water mass.

During the periods of northerly winds, the meridional volume flux towards north below 70 m reached 10 km$^{-3}$ d$^{-1}$ while
southerlies slowed down or even reversed the deep water transport. We estimated the deep water volume flux towards the



north in the order of 400 km$^3$ y$^{-1}$, which is almost one-third of deep layer water mass volume (water below 70 m depth is considered) in the Northern Baltic Proper, Gulf of Finland and Western Gotland Basin. This suggests that water exchange in the deep layers of these basins could occur within 3 years. A longer deep water propagation time scale through the framework of water age has been estimated (Meier, 2007). One reason for the latter discrepancy could be the vertical mixing
of deep water with older water masses, which is taken into account in the study by Meier (2007). Elken (1996) has estimated the transport of 350-400 km$^3$ y$^{-1}$ at the same transect. Our estimation of 400 km$^3$ y$^{-1}$ could somewhat underestimate the transport as the current in the deepest part of the water column was steered by bathymetry. For instance, the thalweg of the Fårö sill is WNW-SSE oriented, which means that the meridional component of the transport there was lower compared to the along-thalweg component. However, uncertainty arising from the local position of the moorings is in place of these
estimations anyhow. For example, there is a strong velocity gradient in the Fårö sill (see Fig. 13). Future papers with numerical simulations, validated with the current measurements presented in this study, are foreseen and should give estimations of transport and fluxes with higher confidence.

The meridional volume flux of 400 km$^3$ y$^{-1}$ and salt flux of 4-5•10$^9$ t y$^{-1}$ towards the north occurred from mid-May to mid-
September 2022 and from mid-January to May 2023, while the net transport was slightly negative from October to mid-January. This seasonal course most probably resulted from the typical seasonality in the wind regime (Soomere and Keevallik, 2001). We can conclude that deep water transport and renewal in the Northern Baltic Proper, Gulf of Finland, and Western Gotland Basin occurs mostly in spring and summer seasons.

The mean cumulative meridional wind stress from April to September was approximately 1 N m$^{-2}$ d$^{-1}$, which is close to the long-term (1983-2022) mean in the same period. Likely, the northward transport of deep water was even more intense in the years when the meridional wind stress was even smaller: the most prominent years were 1990 (-1.5 N m$^{-2}$ d$^{-1}$), 1993-1996 (-1 to -0.5 N m$^{-2}$ d$^{-1}$) and 1997 (-2.6 N m$^{-2}$ d$^{-1}$). In contrast, the wind worked against the deep water northward propagation in 1999 (3.1 N m$^{-2}$ d$^{-1}$), 2004-2005 (2.7-2.9 N m$^{-2}$ d$^{-1}$), 2011 (3.4 N m$^{-2}$ d$^{-1}$), 2012 (4.3 N m$^{-2}$ d$^{-1}$) and 2015 (3.7 N m$^{-2}$ d$^{-1}$).

The impact of the deep water flux on the conditions in the Northern Baltic Proper, Gulf of Finland, and Western Gotland Basin depends on the water properties below the halocline down to the sill depth (Liblik et al., 2018). If water is oxygenated at the sill depth it would rather cause the import of oxygen to the deep layers of these sub-basins. Such a situation occurred from the end of the 1980s to the mid-1990s (Nehring et al., 1994; Rolff et al., 2022), but after the mid-1990s meridional
transport towards north has rather worsened oxygen conditions in these three basins (Liblik et al., 2018; Meyer et al., 2018).

Meridional transport of oxygen-depleted water intrudes to the Gulf of Finland, Northern Baltic Proper, and Western Gotland Basin mostly during the spring and summer months. Moreover, the transport brings denser water to these basins and increases stratification. Seasonal stratification and local oxygen consumption is also higher during the same period



(Stoicescu et al., 2019). All these processes are in superposition in summer and support deoxygenation of the deep layers, and as a result oxygen depletion is strongest in these basins in late summer, early autumn (Lehtoranta et al., 2017).

Interannual variability of wind and meridional transport of oxygen-depleted water alter year to year deep layer water properties in the three basins. For instance, oxygen-depleted areas were more widespread during the years 2013-2014
compared to 2011-2012 (Hansson et al., 2017), which follows the meridional wind stress estimates as shown before.

The vertical structure of the current was strongly linked to the pycnoclines as noted in earlier studies (Bulczak et al., 2016; Lappe and Umlauf, 2016; Paka et al., 2019; Suhhova et al., 2018). The energy at BSD was highest in the upper layer and was dispersed by the pycnoclines. During the period without seasonal stratification higher energy at the BSD reached deeper.
Thus, the occurrence and strength of pycnoclines alter the wind-generated energy transfer at the BSD from the surface to deeper layers.

The kinetic energy in most of the stations/depths could be largely (at least 70%) explained by variability in the LF band (>36 h), similarly as noted earlier in the Gulf of Finland (Lilover et al., 2011). Particularly high, mostly > 80%, was the kinetic
energy share in the cold intermediate layer and in the deep layer. Exceptions were the stations where the seabed depth was in the range of halocline depth. The kinetic energy at the BSD band was much higher in these locations and LF contribution was only 40-53%. Elevated kinetic energy in the halocline near the lateral slope has been explained by the superposition of near-inertial waves and sub-inertial currents (Van Der Lee and Umlauf, 2011). It has been suggested that the closer the halocline is to the seafloor, the stronger the mixing and the oxygen flux is (Holtermann et al., 2022). Processes that occur in
larger spatiotemporal scale, such as mesoscale eddies could break at the boundary to smaller, submesoscale features (Salm et al., 2023; Väli et al., 2017, 2024). An in-depth study about kinetic energy budget, including the energy cascade at the boundary combining ADCP, glider data and high-resolution numerical modelling is planned in the near future in the CABLE consortia.





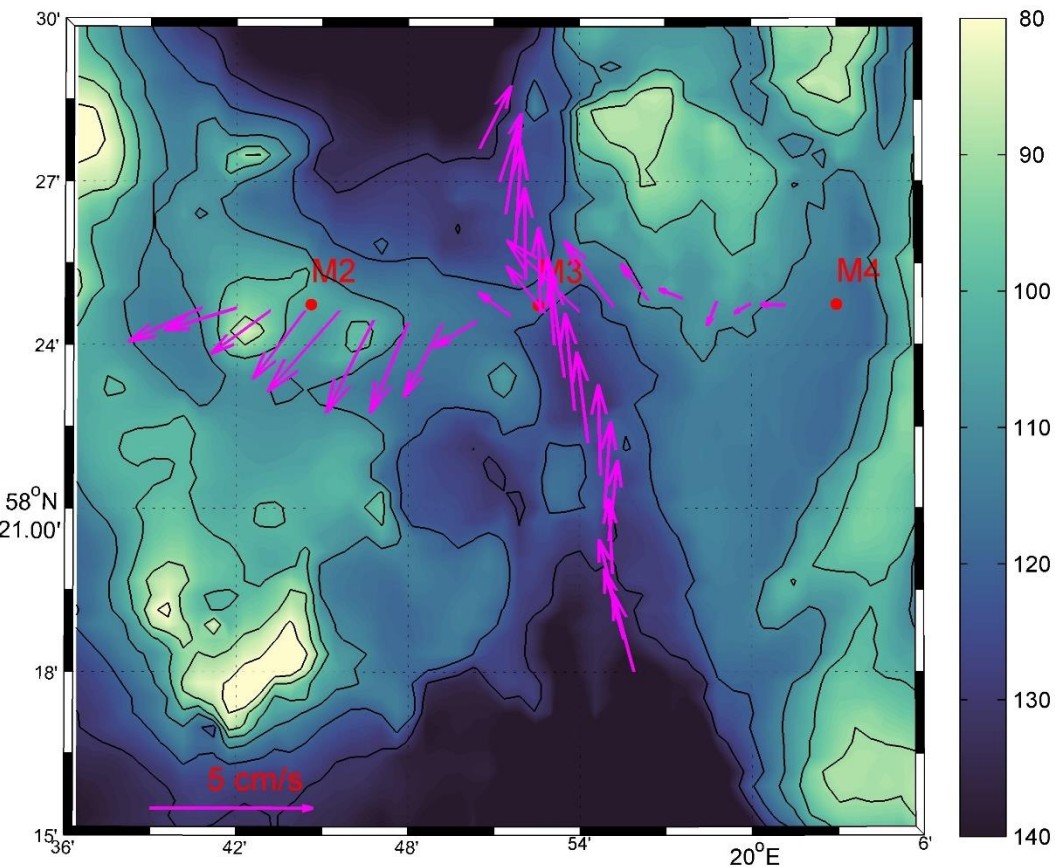

**Fig. 13. Mean current velocity vectors from glider measurements at the layer below 70 m depth from April 2022 to March 2023. Depth (m) is shown in the color scale.**

## 5 Conclusions

The first results of the CABLE study were presented. The circulation and current structure from the surface to the bottom layer vary in synoptic, seasonal, and interannual timescale due to changes in the forcing.

The seasonal mean cyclonic circulation gyre in the upper layer was encouraged by southerly winds and reversed by northerly winds. Northward transport in the deep layer was intensified by northerly winds, and blocked or reversed by southerly winds.



Annual deep water transport of 400 km$^3$ y$^{-1}$ to the Northern Baltic Proper mostly occurs during spring and summertime. This water is denser and oxygen-depleted and therefore increases stratification and worsens the oxygen conditions in the Northern Baltic Proper, Gulf of Finland, and Western Gotland Basin. Changes in the wind forcing alter the interannual variability of this transport.

Stronger currents with higher persistency and higher contribution of LF kinetic energy were observed near the coast while
weaker currents with lower persistency were observed offshore. Ekman spiral was more pronounced in offshore areas while geostrophic current resulting from the sea level gradient prevailed closer to shore. Strong, high persistent gravity current was observed in the near bottom layer of the Fårö sill.

The kinetic energy was largely explained by variability in the LF band (>36 h). The kinetic energy in shorter timescales was
higher in the areas, where the halocline was close to the seafloor.

The next step for the CABLE consortia is to involve numerical simulations, validated with presented measurements, in the analysis. Three main research directions are foreseen. First, transport and flux estimates with higher confidence will be given. Second, a more in-depth analysis of the kinetic energy budget and its spatiotemporal variability will be conducted.
Third, processes in the Fårö sill overflow will be studied in more detail by utilizing ADCP and glider data, and high resolution modelling.

**Competing interests**

The contact author has declared that none of the authors has any competing interests

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
