# Peer review of "Current structure, circulation and transport in the Central Baltic Sea observed by array of moorings and gliders"

_EGUsphere, 2024_

## Author Comment (AC1)

Reviewer 1

The manuscript "Current structure, circulation and transport in the Central Baltic Sea observed by array of moorings and gliders" is a description of one year of mooring and glider observations in the Central Baltic across roughly 58.4N. The paper is descriptive of the physical oceanography of the region. This domain displays estuarine dynamics since it is a shallow (~250 m or less), semi-enclosed, saltwater body.

Reply: Thank you for the review!

Abstract:

-combine                    to              make            one            paragraph
Action: Done.

- "changes in water column habitats" is mentioned in the abstract but not discussed in the text. please remove from the abstract

Reply: We agree, this could be changed.

Action: We changed to "water column characteristics".

- The study "addresses the knowledge gap regarding the current structure and circulation in the Central Baltic Sea." The authors need to provide a better scientific background and reason for completing the study. In the last paragraph of the introduction, the authors describe 4 specific scientific goals of the study. Please describe why you are studying these four items into the first            paragraph            of              the              introduction.
Reply: The text before the 4 specific aims explained the knowledge gaps. We agree, the reasons of the study could be better highlighted.

Action: In the revised version, we conclude the motivations of the study right before the four aims are listed.

- Please remove the sentence "On the one hand, the pycnoclines determine the current shear maxima, but on the other hand, the current structure shapes the pycnoclines" - this statement is vague and not necessary in the introduction.

Reply: We agree

Action: Removed

- The introduction would benefit from a schematic showing what is known about the Central Baltic from modeling studies (and some observations). It would be relevant to the paper to synthesize what is known for the specific study region of the paper.

Reply: We think it is good idea. However, we are afraid the limit of figures is already full. We could try to present what you suggested as a highlight figure that appears next to abstract in web. This figure would include our results as well.

Action: We try to include what you suggested in the so-called highlight figure (graphical abstract).

- Similarly, the paper does not label the Faro Sill on the maps in a way that is clear to me. Please show me the location and bathymetry of Faro Sill. (Could be a part of the schematic)
Reply: We agree it could be presented better.

Action: In the revised manuscript, we show the bathymetry better in Fig. 1. The M3 station was in the Faro sill. We mention that now in several places in the modified manuscript. Also, we added the label to Fig 1.

- On figure 1, what is FD, ND, GD?

Reply: Färö Deep, Northern Deep and Gotland Deep

Action: We added information to the figure caption.

Methods:
- Please merge the paragraphs that are only one sentence.

Action: We merged.

Results:

- As part of figure 2, include a plot of wind direction

Reply: We believe wind stress components already reveal enough information about the wind direction. We are afraid adding another line to the graph would make it difficult to read.

Action: No action.

- Line 216 - since you describe there being 3 water masses, you should plot the three water masses on a T-S diagram

Reply: We agree, it would be nice to show it. But it is technically difficult to include it in the existing figures, and it would not be beneficial enough to create a new figure dedicated to this. We think it is worth citing the T-S diagram figure in Kõuts and Omstedt (1993) paper.

Action: We added citing to the Kõuts and Omstedt (1993) paper.

- In figure 2, what glider observations were used to make the temperature, salinity, and N^2 plots? All the data? How are they averaged? (in space? in time?)
Reply: Yes, it is all the data, averaged daily. There was a sentence about that in the methods chapter, but it could be that this information was difficult to link to Fig. 2.

Action: To avoid this confusion, we cite now Fig. 2 already in the methodology chapter. "Daily mean temperature and salinity profiles from glider data were calculated and presented in Fig. 2."

- Line 226 - A three layer pattern due to the plots of Fig 3 and 4 is not clear. 1) the plots do not show full water column ADCP velocity for most of the mooring sites 2) There is variable bottom depth of the different mooring locations compared to the glider locations to which you are comparing. 3) M5 shows no high shear squared region. I suggest removing this analysis because I do not think that it is strong.

Reply: It was written, "Mostly, a three-layer current structure was present during the period with seasonal stratification." We agree with your three comments, but we believe the cited statement still holds true. Elevated shear was evident around the depth of the pycnocline in most of the stations.

Action: We tried to clarify the statements by adding that we mean deep enough areas only. We also added a sentence about the M5 exception.

- Line 228 - You state that the current is weaker offshore, but this is not true at M2 and M3 where the current is stronger compared to M4 and M5.

Reply: Yes, that is correct and exactly what we mean. M2 and M3 are closer to the boundary compared to M4 and M5.

Action: We rephrased it to avoid confusion: The current was stronger near the boundaries and weaker in the middle part of the section.

- Line 249 - change peed to "speed"
Action: Fixed.

- Line 250 - Since you compare the speeds of the mean profiles, also plot the standard deviation of the mean speed profiles in figure 5.

Reply: We believe, keeping in mind the aims of the paper, that the persistency parameter is enough to describe the variability of current velocities.

- figure 9 - Clearly label the figure's color bar limits that describe the value of correlation. Alternately, make the colorbar go from zero to one.
 Action: We improved the figure according to your first suggestion.

- Figure 12 - is there a weighted mean employed to achieve the plots from the glider observations? Describe in the methods if too much detail to put in the figure caption.

Reply: Arithmetic mean was used in calculations. Horizontal resolution was 0.01° (0.005°) along longitude (latitude) and vertical resolution of 1 m was used.

Action: We added this information to the figure caption.

Conclusions:

- Move the first sentence "The first results of the CABLE study were presented" to the last paragraph in the conclusion.

Action: We did as suggested.

---

## Author Comment (AC2)

Reviewer 3

This manuscript describes the first analysis of an extensive set of observations in the Baltic Sea. Several ADCP mooring and observations from long term glider deployment. The observations is much needed to improve knowledge of the Baltic Sea oceanography.

I am a bit divided on my impression on the manuscript. On the one hand it is based on an new extensive set of observations that is/will be very useful, The analysis is comprehensive and detailed (although sometimes difficult to follow). On the other side, the main impression is that the manuscript is fragmented without a clear thread. After re-reading the manuscript several times I still am uncertain what I have learned form the manuscript, and the message remains unclear.

Reply: Thank you for the time and review! Indeed, the present study is the first general and descriptive investigation of circulation and current structure based on the CABLE data, but further studies focused on various topics will follow.

Action: We have modified the manuscript according to the suggestions of all three reviewers and hope it is better readable now.

My suggestion is that the authors try to include more discussion on results in section 3 Results. I am aware of different views on how to divide results and discussion on results. But as of now I find that the description/discussion on figures are not very enlightening and does not provide much guidance what figures shows, and how they can interpreted. I find section 3.2 (which is a key section) rather difficult to read and to extract useful information on how the system works/operates. Much information is provided, but very little guidance for the reader. (There is a useful discussion on some parts, but in general I find it difficult to get a clear idea how results compare with other studies and what I can learn from present study)-

Reply: Similar study has not been done on the basis of measurements in the Central Baltic. But the results from numerical models could be discussed.

Action: We tried to include some information from earlier studies to highlight better the results of the current paper. We could not add too many details, as it would repeat the discussion and the results part would be too long. We also modified the introduction to better justify the aims and reasons of the current study.

Major specific points.

Avoid one sentence paragraphs. Try also to avoid paragraphs with few sentences. It gives a fragmented impression of the study.

Reply: We agree.

Action: We changed that in the revised version.

I notice that there are no comparison with results from other studies in the results section. This implies that it is rather difficult to get a clear picture how important the result from the study is, does it agree or disagree with earlier studied. Including some discussion in result section would improve readability ad impact of the study.

Reply: As we mention above, we tried to include some results from previous studies.

Figures are difficult to read:

Figure 1: The stations are difficult to read in the plot.

Reply: We agree.

Action: We improved Fig. 1.

Figures 2, 3, 9: The colorbars are too small to read (at least on paper where you cannot zoom in).

Action: We made the colorbars larger.

Figures 6, 9: The arrows seem to describe current speed and direction. Please clarify what is shown.

Action: We clarified it in the Fig. 6 caption a bit more. These are mean vertical profiles of current velocity, i.e. speed and direction are combined. Fig. 9 shows the angles between the current vector and the wind, and the correlation strength between current and wind velocities.

Figure 6: What is x-axis (guess stations but ...).

Reply: Yes, these are stations.

Action: We added labels to x-axis.

Figure 3, 4 are hardly discussed but take up significant space. Please provide a relevant discussion. It is not clear to what the purpose what the purpose of Figure 4. Would readability increase with a log colorscale?

Reply: The main purpose of Figures 3 and 4 is to present the time series that we analyze later. We agree, the figures are not very thoroughly explained and cited, but we wanted to avoid long descriptions of time series.

Action: In the revised version, we use Figs. 3-4 a bit more, but we still prefer to avoid long descriptions about these time series.

Open access to data. I understand that the team wants to use the observation for more studies. It is stated that observations will be used for e.g. validating ocean models etc. It would be great to write out if data will be publicly available at some point (in, say two years).

Reply: We indeed plan to use the data together with numerical modelling to give better transport estimates in our next study. That would include validation of the model as well. We believe that the data can be published together with our next paper.

Action: We will write about the data availability.

Minor specific points

Line 115. How many time series and at what depth? (mentioned later, but should be clarified here).

Action: We added details there.

Table 1: Table should be adjusted to show date in clearer way.

Reply: We agree.

Action: We adjusted it.

Table 1: Define what EMDB293, any reference to cruise report (assuming it is a cruise).

Action: We decided to remove these.

Line 157: Define what IGRF stands for.

Action: We added the definition.

Line 158: Not sure what 2-sided derivative algorithm is. Do you mean central numerical scheme. Please clarify.

Reply: Yes, we meant central differencing scheme.

Action: We modified the text accordingly.

Line 187 and following paragraph. It is difficult to read. Perhaps using bullet points improve readability.

Action: We use bullet points in the revised version.

Line 207: High P (79-97%) of wind. It is unclear to me, do you mean that wind has high persistence?

Reply: Yes, we mean exactly that.

Action: We changed the sentence to make it more clear: "Periods were characterized by high persistency (P = 79-97%) of wind."

Figure 2: I do not see the point in having the wind speed "attached" to this figure. It would connect better to figure 3.

Reply: We more or less agree with this, but for technical reasons, we prefer to keep it as it is. Fig. 3 is already very large.

Line 249: Spelling peed.

Action: Fixed.

Line 250: "acoustic measurements". I presume it is ADCP measurements.

Action: We changed acoustic to ADCP.

Line 291. You mention "diurnal tidal constitutents", but could not 24 h period reflect daily cycle (in, e.g. wind forcing etc).

Reply: You are right.

Action: We added "diurnal cycle" to the text.

Figure 7: It would help if the most important time "periods" is illustrated in the figure (e.g as lines/tics in the "x-axis".

Reply: We agree it would be useful.

Action: We marked frequencies corresponding to 17-h and 36-h periods to in the figure.

Line 414: Please explain how the volume fluxes are consistent with cumulative wind stresses. What assumptions do you make, and what result do you get? Please enlighten the reader.

Reply: These explanations have been given in chapters 3.3 and 3.4. In the previous paragraph, we remind the reader about the strong correlation between wind and sub-halocline meridional current.

"On one hand, considerably high correlation strength ρ was noted between current in the deep layer at stations M3, M4, M6, and M8. On the other hand, the deep layer flow towards the north at these stations was significantly correlated to the wind from northerly directions (Fig. 9). As a result, one could see two events created by southwesterly wind impulses (July and late September – early October) when the meridional current was directed southward at all mentioned stations (Fig. 11a)."

Action: We complemented the section you cite to make it more clear.

"Despite short-term reversals, the cumulative meridional transport towards the north was approximately 270 km3 from mid-May to mid-September 2022, while it was slightly negative from October 2022 to mid-January 2023 and 140 km3 from mid-January to mid-March 2023. These volume fluxes were in accordance with cumulative meridional wind stress, which was approximately 1 N m-2 d, 6 N m-2 d, and -1 m-2 d, respectively during the same periods (Fig. 2a). Thus, during the periods (seasons), when meridional wind stress was southward or close to neutral, the meridional transport towards the north was approximately 70 km3 month-1 while the sub-halocline meridional transport was halted during the period of high positive meridional wind stress."

---

## Author Comment (AC3)

Reviewer 2

The authors investigate the structure and properties of the current field in the Central Baltic Sea using current data from an array of eight moorings in a line at the latitude of the Fårö Sill, seven equipped with a profiling ADCP and one with a classical current meter at a fixed level. These data were supplemented with ERA5 reanalysis wind data to take the forcing into account as well as some bottom temperature and salinity data from three of the moorings and CTD data from several glider surveys in that region to describe the hydrographical situation and calculate salt transports.

In detail, the authors present and examine stratification, current shear, profiles of mean current speeds and persistency, the seasonal variation of current velocity profiles, i.e. dependence on seasonal stratification, profiles of HKE spectra and profiles of HKE content in specific frequency ranges and their share in total HKE, profiles of complex correlation (magnitude and phase) between current and wind, profiles of mean current vectors in periods of nearly constant wind forcing, i.e. dependence on wind direction, to characterise the current field. Additionally, they calculate volume and salt transports across the line of moorings below 70 m and show temperature and salinity sections and mean velocities from glider sections along and across Fårö Sill to illustrate the overflow over the sill.

With this investigation the authors provide some substantial facts on the basis of measurements beyond the existing knowledge from simulations, while the presentation of the results and in particular their discussion unfortunately remain largely rather descriptive than quantitatively explanatory. However, the manuscript covers the primary aims of this work given by the authors in the introduction, which are all of more or less descriptive nature, very well.

The language is good and reads fluently. Some figures would benefit from minor improvements, see specific comments.

Reply: Thank you for your time and suggestions! Indeed, the present study is the first general, descriptive investigation of circulation and current structure based on the CABLE data, but further studies focused on various topics will follow. We modified the manuscript according to the specific comments.

Specific comments:

Figure 1: The coloured and dashed lines in the left map are hardly to distinguish. I suggest to rework this figure to make it clearer.

Reply: We agree it could be improved.

Action: We reworked the figure.

line 128: I think it would be good to give also the spatial resolution of the ERA5 data here, i.e. the area for which the used grid point is meant to be representative for.

Reply: We agree.

Action: We added information about the spatial resolution in the text.

Table 1: From Figures 3 and 4, I guess the starting day of the deployment period of M2 should be somewhat later than 09 May, which seems to be correct for M3.

Reply: Indeed, there was an error in the table. Thank you for noticing!

Action: We fixed the table.

line 154: This has to be lined out more explicitly. Why are the data low-pass filtered in general? What has been done exactly, in particular, for the calculations of HKE? For example, if the data is low-pass filtered, miscalculations of HKE in particular in the BSD are the consequence. Here you state mainly inertial oscillations, tides and seiches are suppressed by your filtering. In line 180 it says these are the main contributors to the BSD. This confuses the reader. I am sure this is not what you did. Please elaborate your filtering (What is done for which purpose and calculation?) in more detail to avoid confusion.

Reply: You are right, the sentence about filtering with a 36-hour cut-off filter is misplaced and confusing. 36-h is used only for Fig. 3 and Fig. 4. Otherwise, 1-h values are used, including for the spectrum and HKE.

Action: We modified the text accordingly to avoid the confusion.

line 184: I think it is useful for the less experienced readers to explicitly give the definition of $N^2$.

Reply: we agree.

Action: We added the definition.

line 190: 'zonal' -instead of 'meridional'

Action: We fixed it.

line 190: 'black' instead of 'dashed', see text under Figure 1 and take account of the comment on it.

Action: We fixed it.

line 196: The regression of the volume transport with respect to the near bottom meridional velocity at M3 can be justified assuming a similar (meridional) current distribution over the considered transect cross-section. In addition, the same regression of the salt transport needs the assumption of similar distribution of salinity over the transect cross-section. Can you show to which extend these assumptions are valid and estimate the errors which are introduced by the supposed deviations from the assumptions?

Reply: We have shown the time series of the regression-based estimates and the whole section based estimates in Fig. 11. You are correct about the assumption of a similar distribution of salinity. Horizontal salinity gradient along the zonal transect is minor. Especially, compared to meridional and vertical salinity gradients. We also made measurements of water column properties in the zonal section and in the section from Gotland Deep to the Gulf of Finland. Our conclusion is that some errors result from the assumption of even salinity at the section,

but the major source of errors is the coarse array of moorings. We show the measured distributions in October 2022 below. We agree that these assumptions might be more highlighted in the manuscript.

[Figure]

Fig. 1. Section from the Gotland Deep to the Gulf of Finland

[Figure]

Fig. 2. The zonal section.

Action: We now draw attention to assumptions and potential errors both in the Data and Methods chapter and in the Discussion.

line 250: This statement is certainly correct also for M10, but not very meaningful for the point measurement there. As it is also not an acoustic measurement and therefore implicitly excluded from the list in line 249, I would simply remove it explicitly from the list in line 249.

Reply: Indeed. Thank you for noticing.

Action: We removed it.

line 262: According to Figure 5, the persistency at 58 m in M9 is rather 54 % than 61 % to me.

Reply: Indeed. Thank you for noticing.

Action: We fixed it.

line 263: According to Figure 5, the persistency at the thermocline in M9 is rather 49 % than 51 % to me.

Reply: Indeed. Thank you for noticing.

Action: We fixed it.

line 268: I think I know what you mean, but for better reading you should briefly explain in which respect the cyclonic circulation is reflected by the mean velocity profiles. Else, for some readers, it may be hard to understand what exactly you mean here.

Reply: We agree. It could be a bit more described.

Action: We added a sentence about mean velocities in the eastern and western part of the section.

line 269: I am afraid I have a problem with the wording here. What is the difference between 'interior basin' and 'central part'? Do you mean 'basin rim' and 'basin centre', respectively? Please clarify this wording.

Reply: We agree, there was confusion.

Action: We changed the sentence: „The mean flow was stronger closer to the boundary and weaker in the central part of the basin."

line 303: Beside the technical correction to this line, I have a problem to understand what you want to say with this sentence. Do you mean something like: 'A somewhat higher energy at the bottom in the BSD and BD bands was revealed at the two stations where the sea depth was in the range of the halocline depth (M2, M9).' This is what I interpret. Please clarify this.

Reply: Your interpretation was what we meant.

Action: We replaced the sentence with the one you wrote. Thank you.

line 305: Similar to the preceding comment, this sentence would make much more sense to me if it started, for example, with 'A significantly higher energy …' in addition to the change proposed in the respective technical correction.

Action: We made the change.

Figure 7: In the text to the figure it says in brackets that HKE spectra equal HKE spectral density multiplied by the frequency. I would rather expect that HKE spectra equal HKE spectral densities multiplied by the used frequency step or interval like a finite integration. Please check this and correct if necessary.

Reply: Indeed, we use here the product of the spectral density and the corresponding frequency. This procedure reduces the slope of the spectrum, which is important for better visualization. Furthermore, the so-called variance-preserving spectra preserve the signal variance under the spectral curve (Emery and Thomson, 2004).

line 328: This is quite a simplification. Correct is that the vertical integrated transport is to the right of the wind vector. So, this should be reformulated somehow.

Reply: We corrected it.

Action: It now reads, "the transport in the upper layer was to the right from the wind vector".

line 338: This is not directly visible from Figure 9 as the depicted vector sticks only show the relative angle between current and wind. Therefore, this statement should be explained some more.

Reply: We agree it needed more explanation.

Action: In the revised version, we first mention that the best correlation with the deep layer current was with the wind from the opposite direction.

Figure 9: In the first sentence, I suggest to replace 'current direction' and 'wind direction' by 'current vector' and 'wind vector' as the wording is in the second sentence to avoid confusion as the wind direction is opposite to the wind vector. Furthermore, I would add 'α' to 'mean angles' and 'correlation strength' to the variable name 'ρ'.

Reply: Good suggestions. Thank you.

Action: We changed as suggested.

Figure 9: I am not sure whether the information of plots would be better or easier to get if the vector sticks are coloured instead of an extra row of dots with the colour information. Maybe the correlation strength could also be shown as the length of the vector sticks if a suitable scale can be found, or a combination of both, i.e. coloured vector sticks with variable length. I think it is worth to try this out.

Reply: Good suggestions. We tried both, but it seems the current version works best still.

Action: We removed every second vector for better readability of the figure though.

line 402: Like in Figure 9, I would add 'correlation strength' to the variable name 'ρ'.

Action: We added it.

Line 415: Unit of cumulative wind stress should be [N m$^{-2}$ d] instead of [N m$^{-2}$ d$^{-1}$] like in Figure 2.

Action: We fixed it.

line 422: According to section 2.2. it should be 22 April instead of 24 April and the link is somewhat different there. Please equalise both or clarify.

Action: We equalised it.

line 427: How does this relate to the Baltic residence times of about 30 years give elsewhere in literature? Is it an extraordinary high transport to that area observed in that year or does the water reside somewhere else before entering of after leaving that region for the rest of the time?

Reply: It was not extraordinarily high. Elken (1996) had similar transport estimates based on temperature-salinity data. Likewise, the wind stress in the particular year was similar to the long-term mean. We have dealt with the topic in discussion.

Action: We added another sentence to the discussion to point out that the particular year was likely close to the long-term mean in terms of subhalocline transport.

line 483: Why was something similar in contrast possible for the surface layer in Section 3.4., lines 360 to 362? What is the difference in consideration and interpretation?

Reply: Line 483 described one-year mean current vectors > 70 m in the sill. Lines 360-362 described the situation during SW wind prevailing in the upper layer.

line 502: I suggest to write 'southwesterly' instead of 'southerly and westerly', because that is what was investigated.

Action: We changed it accordingly.

Line 511: Like before, I suggest to write 'northeasterly' instead of 'northerly', because that is what was investigated.

Action: We changed it accordingly.

line 538: Should be 'NNW-SSE' instead of 'WNW-SSE' I guess, because that are opposite directions and the resulting orientation fits.

Action: We fixed it.

line 538: This statement is wrong what can easily be seen from Gauss's Law as we certainly have a divergence-free current field. Or in other words, the higher current velocity along the channel would be exactly compensated by the smaller cross-section perpendicular to it in comparison to the larger zonal cross-section in combination with the smaller meridional current velocity.

Action: We removed that part.

line 597: I do not see a reversal of the cyclonic circulation in the upper layer in the results presented in section 3.4. for the first (southwesterly wind) and the second (north-northeasterly wind) period considered. The current reverses from north to south in the eastern part of the array from the first to the second period. But in the in the west at M2 in both periods the current is to the west.

Reply: You are right, it cannot be stated like that on the basis of our mooring data. We suspect it is related to the fact that there is an open boundary in the west in the upper layer in the location of M2, which means that the signs of reversal likely could be seen along the mainland of Sweden, i.e. out of our array.

Action: We modified the sentence in a way that it only is about the eastern boundary.

technical corrections:

line 106: '… deployed at the same …' instead of '… deployed to the same …'

line 115: '… were recorded at …' instead of '… was recorded in …'

line 163: Equation 1: I suggest to remove '/n' in the denominator and to put the fraction '1/n' in front of the summation in the denominator instead for better readability.

line 171: Emery and Thomson (2004) is missing in the references.

line223: add unit 'm' to '60-80'

Figure 4: add unit $(1/s^2)$ to the colourbar.

line 249: typo 'speed' not 'peed'

line 298: better: The HKE share of the BD band at the Fårö Sill … , but the HKE in the LF band

line 300: '… energy in the LF band …' instead of '… energy at the LF band …'

line 301: '… energy in the BSD and BD bands in …' instead of '… energy at BSD and BD in …'

line 303: '… energy in the BSH and BD bands was …' instead of '… energy at the BSH and BD was …'

line 305: '… energy in the BSD band was also revealed …' instead of '… energy at the BSD was revealed also …'

line 376: better to understand and less irritating: 'vice versa' instead of 'opposite'

Figure 12: Exchange '0.1 g kg$^{-1}$' and '0.1 °C' to make their order respective to 'temperature and salinity' before.

Reply: Thank you for the help!

Action: We fixed all.